# Comparative Analysis of Myokines and Bone Metabolism Markers in Prepubertal Vegetarian and Omnivorous Children

**DOI:** 10.3390/nu16132009

**Published:** 2024-06-25

**Authors:** Jadwiga Ambroszkiewicz, Joanna Gajewska, Katarzyna Szamotulska, Grażyna Rowicka, Witold Klemarczyk, Małgorzata Strucińska, Magdalena Chełchowska

**Affiliations:** 1Department of Screening Tests and Metabolic Diagnostics, Institute of Mother and Child, Kasprzaka 17A, 01-211 Warsaw, Poland; joanna.gajewska@imid.med.pl (J.G.); magdalena.chelchowska@imid.med.pl (M.C.); 2Department of Epidemiology and Biostatistics, Institute of Mother and Child, Kasprzaka 17A, 01-211 Warsaw, Poland; katarzyna.szamotulska@imid.med.pl; 3Department of Nutrition, Institute of Mother and Child, Kasprzaka 17A, 01-211 Warsaw, Poland; grazyna.rowicka@imid.med.pl (G.R.); witold.klemarczyk@imid.med.pl (W.K.); malgorzata.strucinska@imid.med.pl (M.S.)

**Keywords:** myokines, bone metabolism markers, vegetarian diet, prepubertal children

## Abstract

The role of bone and muscle as endocrine organs may be important contributing factors for children’s growth and development. Myokines, secreted by muscle cells, play a role in regulating bone metabolism, either directly or indirectly. Conversely, markers of bone metabolism, reflecting the balance between bone formation and bone resorption, can also influence myokine secretion. This study investigated a panel of serum myokines and their relationships with bone metabolism markers in children following vegetarian and omnivorous diets. A cohort of sixty-eight healthy prepubertal children, comprising 44 vegetarians and 24 omnivores, participated in this study. Anthropometric measurements, dietary assessments, and biochemical analyses were conducted. To evaluate the serum concentrations of bone markers and myokines, an enzyme-linked immunosorbent assay (ELISA) was used. The studied children did not differ regarding their serum myokine levels, except for a higher concentration of decorin in the vegetarian group (*p* = 0.020). The vegetarians demonstrated distinct pattern of bone metabolism markers compared to the omnivores, with lower levels of *N*-terminal propeptide of type I procollagen (P1NP) (*p* = 0.001) and elevated levels of *C*-terminal telopeptide of type I collagen (CTX-I) (*p* = 0.018). Consequently, the P1NP/CTX-I ratio was significantly decreased in the vegetarians. The children following a vegetarian diet showed impaired bone metabolism with reduced bone formation and increased bone resorption. Higher levels of decorin, a myokine involved in collagen fibrillogenesis and essential for tissue structure and function, may suggest a potential compensatory mechanism contributing to maintaining bone homeostasis in vegetarians. The observed significant positive correlations between myostatin and bone metabolism markers, including P1NP and soluble receptor activator of nuclear factor kappa-B ligand (sRANKL), suggest an interplay between muscle and bone metabolism, potentially through the RANK/RANKL/OPG signaling pathway.

## 1. Introduction

Bone and muscle interact both anatomically and mechanically, and through paracrine and endocrine signals [1,2,3,4]. They produce and release hormonal factors that mutually influence each other, contributing to bone–muscle crosstalk [5,6,7].

The assessment of bone turnover relies on biochemical markers reflecting the dynamic balance between bone formation and resorption processes [8,9]. Among them, *N*-terminal propeptide of type I procollagen (P1NP) is a marker of the early stage of bone formation, synthesized during type I collagen production, the primary constituent of bone matrix. Another bone formation marker, osteocalcin (OC), is a protein synthesized by osteoblasts involved in bone mineralization. A bone resorption marker that can be determined in the serum is the *C*-terminal telopeptide of type I collagen (CTX). It is released during the breakdown of type I collagen and reflects the activity of osteoclasts, the cells responsible for bone resorption [10,11].

The balance between bone formation and resorption is regulated by various factors, including the RANKL/RANK/OPG and Wnt/β-catenin signaling pathways [12,13]. Receptor activator of nuclear factor kappa-B ligand (RANKL) is a cytokine essential for osteoclast formation and activation. It binds to its receptor (RANK) on osteoclast precursors, promoting their differentiation into mature osteoclasts and stimulating bone resorption. Osteoprotegerin (OPG) acts as a RANKL decoy receptor, inhibiting its binding to the receptor on osteoclasts and thereby reducing bone resorption [14]. Sclerostin, a protein secreted mainly by osteocytes, is recognized as a negative regulator of bone growth due to its role as an inhibitor of the Wnt/β-catenin pathway essential for osteoblast function [15,16].

Recent interest has focused on understanding the interplay between myokines, cytokines secreted by muscle in response to exercise, and bone metabolism [17]. Myostatin (MSTN), a prominent myokine, negatively regulates muscle growth and may impact bone remodeling dynamics. Existing evidence suggests that its elevated levels have been associated with increased bone resorption, decreased bone formation, and, consequently, reduced bone mass [18,19,20,21]. Decorin (DCN), a small leucine-rich proteoglycan secreted by muscle and other tissues, is involved in tissue remodeling, inflammation modulation, and collagen fibrillogenesis [22,23]. Its deficiency results in altered tissue structure and function. Several in vitro and animal studies have shown that decorin exhibits positive effects on bone formation by promoting osteoblast differentiation and also may regulate bone resorption by decreasing osteoclast activity [24]. Myonectin and fibroblast growth factor-21 (FGF-21), linking muscle with glucose and lipid metabolism, also emerge as significant players in metabolic regulation, potentially affecting bone metabolism [25,26]. Although the exact mechanisms underlying these associations remain unclear, it is suggested that decreased levels of these myokines can lead to reduced bone mineral density and impaired bone formation [27,28].

As dietary patterns shift, with more individuals embracing meat-free diets, concerns arise about nutrient adequacy, particularly in children [29,30]. Optimal diets during childhood are critical for achieving an age-appropriate body composition, including muscle and bone mass [31,32,33]. While numerous international health organizations have recommended vegetarian diets as healthy, especially in terms of lower cardiovascular risk, less evidence exists for children [34,35,36]. Given the increasing number of children following unconventional diets, understanding the relationships between myokines and bone metabolism in children following different dietary patterns can offer crucial insights into optimizing nutritional strategies for skeletal health.

This study aims to evaluate the serum concentrations of bone turnover markers and myokines and explore associations between myokines, bone markers, and anthropometric parameters in children following vegetarian and omnivorous diets.

## 2. Materials and Methods

### 2.1. Subjects

This study enrolled a total of 68 prepubertal healthy children (aged 4–9 years) from a group of consecutive patients attending the Institute of Mother and Child (Warsaw, Poland) and was conducted between June 2022 and November 2023. Among the participants, 44 (50% male and 50% female) had followed a vegetarian diet since birth. Of these children, 34 (77%) were lacto-ovo-vegetarians, 4 (9%) were lacto-vegetarians, and 6 (14%) were vegans. The inclusion criteria were being in the prepubertal period, adhering to a vegetarian diet from birth, and being apparently healthy without developmental or nutritional disorders and without any history of bone fractures. Exclusion criteria included low birth weight, gastrointestinal diseases, history of chronic infection, and regular intake of medication (except for standard vitamin D supplementation and vitamin B_12_ supplementation in the case of vegans).

The control group comprised 24 healthy children (48.0% male, 52.0% female) who followed a traditional omnivorous diet, inclusive of meat, poultry, and fish.

The participants’ health status was evaluated through medical history data collection and a basic physical examination, including Tanner stage assessment [37]. The data for physical activity (PA) were obtained from the questionnaires completed by the parents and were found to be similar between vegetarians and omnivores. On average, participants engaged in about 60–90 min per day of moderate-to-vigorous physical activity (MVPA) and approximately 30 min per day of vigorous physical activity (VPA), including regular activities after school. A similar number of children in the vegetarian and omnivorous groups were studied during the autumn–winter period (October–April) and the spring–summer period (May–September) to avoid differences in vitamin D levels caused by varying sunlight exposure across different seasons.

This study adhered to the principles of the Helsinki Declaration and received approval from the Ethics Committee of the Institute of Mother and Child (decision number 15/2022, approval date 5 May 2022). Informed written consent was obtained from the parents of all the participating children before this study commenced.

### 2.2. Anthropometric and Nutritional Measurements

Anthropometric measurements were conducted using a calibrated stadiometer and electronic scale. Body weight and height were taken to calculate body mass index (BMI), obtained by dividing body weight (in kilograms) by height squared (in meters). BMI values were converted to a normalized BMI z-score using age- and sex-specific norms [38].

Dietary assessment was performed using a nutritional software program Dieta5^®^, Extended version Dieta 5.0 (National Food and Nutrition Institute, Warsaw, Poland) [39]. The parents of the children involved in this study were instructed by a nutritionist to keep a food diary for their children, as described in more detail in the previous article [40]. Three dietary recalls (two weekdays and one weekend day) were conducted to evaluate average daily energy intake and the percentage of energy from protein, fat, and carbohydrates [41].

### 2.3. Biochemical Analyses

Venous blood samples were collected in the morning after an overnight fast to minimize diurnal variations. The blood samples were centrifuged at 2500× *g* for 10 min at 4 °C to obtain serum, which was divided into tubes and stored at −80 °C for up two months before undergoing biochemical analysis. Biochemical parameters were measured in all children, except for myonectin and decorin, which was determined in 42 (95%) vegetarian and in 22 (92%) omnivorous subjects.

Serum concentrations of 25-hydroxyvitamin D were determined using an electrochemiluminescent immunoassay (ECLIA) with kits from DiaSorin Inc. (Stillwater, OK, USA) on a Liaison analyzer. The coefficient of variation (CV) for this assay was 6.0–9.8%. Serum concentrations of myokines and bone metabolism markers were determined using commercial enzyme-linked immunosorbent assay (ELISA) kits, following the manufacturer’s instructions. Decorin and myonectin levels were measured using ELK Biotechnology (Wuhan, China) kits. The limit of detection for these methods was 0.54 ng/mL, and the intra-assay and inter-assay CVs were below 8% and 10%, respectively. The concentration of FGF-21 was assessed with the Human Intact FGF-21 kit from Epitope Diagnostics (San Diego, CA, USA), with a limit of detection of 1.7 pg/mL. The intra-assay precision was 4.2–5.7% and the inter-assay precision was 1.9–6.9%. Myostatin levels were assayed using SunRed Biotechnology (Shanghai, China) kits, with a limit of detection of 0.05 ng/mL, and an intra-assay and inter-assay precision of less than 8% and 11%, respectively.

From the bone metabolism markers, the serum concentrations of P1NP, OC, CTX-I, soluble RANKL (sRANKL), OPG, sclerostin, and IGF-I were assessed. Serum P1NP and sRANKL were determined using Human P1NP and Human sRANKL kits from SunRed Biotechnology (Shanghai, China), with a limit of detection of 5.125 ng/mL for P1NP and 1.56 pg/mL for sRANKL. The intra-assay and inter-assay CVs were less than 10% and 12% for P1NP, and less than 9% and 11% for sRANKL, respectively. Serum levels of OC and CTX-I were detected using N-MID Osteocalcin and Serum CrossLaps (CTX-I) kits from Immunodiagnostic Systems (Boldon, UK), with a limit of detection of 0.50 ng/mL for OC and 0.02 ng/mL for CTX-I. The intra-assay and inter-assay CVs were 1.3–2.2% and 2.7–5.1% for OC, and 1.7–3.0% and 2.6–10.9% for CTX-I, respectively. OPG concentrations were assessed using a kit from DRG Instruments GmbH (Marburg, Germany), with a limit of detection of 0.03 pmol/L. The intra-assay CV ranged between 2.5 and 4.9%, and the inter-assay CV ranged between 1.7 and 9.0%. Serum IGF-I was determined using a kit from Mediagnost (Reutlingen, Germany), with an analytical sensitivity of 0.091 ng/mL. The intra-assay CV ranged between 5.08 and 6.65%, and the inter-assay CV ranged between 5.53 and 6.56%. The levels of sclerostin were assessed using the TECO Human Sclerostin HS kit from TECOmedical AG (Sissach, Switzerland), with a limit of detection of 0.058 ng/mL. The intra- and inter-assay CVs were between 1.7 and 8.2% and 1.6 and 4.8%, respectively. Serum IGF-I was determined using a kit from Mediagnost (Reutlingen, Germany), with an analytical sensitivity of 0.091 ng/mL. The intra-assay CV was between 5.08 and 6.65%, and the inter-assay CV was between 5.53 and 6.56%.

### 2.4. Statistical Analyses

The normality of the variables’ distribution was assessed using the Kolmogorov–Smirnov test. Descriptive statistics included mean and standard deviation (SD) for normally distributed variables, and median value and interquartile range (IQR) for non-normally distributed variables. Group comparisons were conducted using exact Mann–Whitney U test and chi-square test. Due to the small size of the control group, correlation analysis was performed only in the vegetarian group using the Spearman test. Multivariate quantile regression was applied to adjust bivariate associations with *p* < 0.1 for age. A *p*-value of <0.05 indicated statistical significance. Statistical analyses were performed using IBM-SPSS software version 23.0 (SPSS Inc., Chicago, IL, USA) and STATA 18 (Stata Corp LLC, College Station, TX, USA).

## 3. Results

All 68 participants, consisting of 44 vegetarians (22 boys, 22 girls) and 24 omnivores (13 boys, 11 girls) were healthy, normal-weight, prepubertal children of Caucasian ethnicity (Table 1).

The groups were comparable in terms of age, sex, and anthropometric parameters. Regarding diet, both groups exhibited a similar total daily energy intake and percentage of energy from fat. However, the children on a vegetarian diet had a significantly higher percentage of energy from carbohydrates (*p* = 0.011) and a lower percentage of energy from protein (*p* < 0.001) compared to the omnivores. Additionally, the dietary intake of protein (in grams per day) was significantly lower (*p* < 0.001) in the children on a vegetarian diet than in the meat-eaters. The dietary calcium, phosphorus, magnesium, and vitamin D intakes did not differ significantly between the two studied groups.

The serum concentrations of biochemical parameters, including bone markers and myokines, in the two studied groups of children are presented in Table 2.

Concerning bone metabolism markers, the vegetarian children exhibited a significantly lower P1NP concentration (*p* = 0.001), but higher levels of CTX-I (*p* = 0.018) compared to the omnivores. Consequently, the ratio of P1NP/CTX-I was decreased in the vegetarians. No significant differences were found in the levels of regulatory markers, IGF-I, or 25-hydroksyvitamin D between the two groups.

Regarding myokines, similar levels of myostatin, myonectin, and FGF-21 were observed in both groups, while the concentration of decorin was significantly higher in the vegetarians (*p* = 0.020).

Analyzing the correlations among bone markers in the vegetarians, we found that the OC level was negatively associated with the P1NP concentrations (rho = −0.458, *p* = 0.002) and the P1NP/CTX-I ratio (rho = −0.474, *p* = 0.001), while it was positively correlated with IGF-I (rho = 0.465, *p* < 0.001). The CTX-I levels were associated with the IGF-I (rho = 0.360, *p* = 0.016) and sclerostin values (rho = 0.389, *p* = 0.009). Additionally, sclerostin was inversely related to the P1NP/CTX-I ratio (rho = −0.354, *p* = 0.018), and IGF-I was weakly correlated with the 25-hydroxyvitamin D concentration (rho = −0.316, *p* = 0.042).

Most of the bone metabolism markers correlated with age, including the following: P1NP (rho = −0.358; *p* = 0.017), OPG (rho = −0.311; *p* = 0.040), sRANKL (rho = −0.322; *p* = 0.033), sclerostin (rho = 0.312; *p* = 0.040), IGF-I (rho = 0.435; *p* = 0.003), 25-hydroxyvitamin D (rho = −0.462; *p* = 0.002).

Regarding the myokines, myostatin, myonectin, and FGF-21 were negatively correlated with age and anthropometric parameters (weight and height), with myostatin showing the strongest associations in the vegetarians (Table 3). Myokines were also related to dietary parameters, with inverse associations of myostatin and FGF-21 with the dietary energy and protein intakes.

The associations between the serum concentrations of the myokines and bone metabolism markers are presented in Table 4. In the vegetarians, significant positive associations were found between the myostatin levels and P1NP, sRANKL, and the P1NP/CTX-I ratio. Additionally, serum myostatin negatively correlated with osteocalcin and the OC/CTX-I ratio. Another myokine, myonectin, was significantly positively correlated with the OPG levels.

The bivariate relationships between myokines and bone metabolism parameters (*p* < 0.1) were adjusted for age using multivariate quantile regression models, in which a single myokine and age were the independent variables and a single bone metabolism parameter was the dependent variable. Only the relationships between the myostatin and P1NP concentrations (*p* < 0.001) and between the myostatin and sRANKL concentrations (*p* = 0.007) remained significant after adjustment (Table 5).

To visualize the relationships described above, and considering that the adjustment for age did not substantially change the β coefficients (by less than 10%), the crude quantile regression lines were plotted within the framework of the original data (see Figure 1 and Figure 2).

## 4. Discussion

The scientific literature offers limited data regarding biochemical bone turnover markers in children following a vegetarian diet. The authors have reported an impaired or similar rate of bone turnover in vegetarians and omnivores [42,43,44]. Additionally, studies have suggested normal or decreased bone mineral density in vegetarians, particularly in vegans [45,46,47,48]. However, comparing these findings is difficult due to variations in study populations, with our investigation focusing on prepubertal children while others predominantly involved adults.

The results of the present study indicate that children adhering to a vegetarian diet exhibit impaired bone metabolism with reduced bone formation activity (as indicated by P1NP) and increased bone resorption (as indicated by CTX-I) compared to those consuming a meat-containing diet. Notably, the concentration of osteocalcin, another bone formation marker, did not differ between the studied groups. This could be explained by the fact that P1NP is a marker of early-stage bone formation (collagen synthesis), whereas osteocalcin, synthesized in osteoblasts, is involved in regulating bone metabolism and bone mineralization. This interpretation is supported by the negative correlations observed between osteocalcin and P1NP, as well as the P1NP/CTX-I ratio. OC also plays a crucial role in glucose metabolism, favoring nutrient update and catabolism in muscle and is necessary for adaptation to exercise [10,49]. Serum OC, the synthesis of which is vitamin K-dependent, exists as the carboxylated (cOC) and undercarboxylated (ucOC) forms [50]. Most studies indicate that the endocrine activity of OC in humans is primary attributed to ucOC [51]. Undercarboxylated osteocalcin, a hormone secreted by bones, may affect energy metabolism and muscle function and is required for exercise adaptation. While our previous study [52] showed a significantly higher cOC/ucOC ratio in vegetarians compared to omnivores, in the current study, we only assessed the total osteocalcin levels and so did not observe differences between groups.

Although no significant differences were found in the OPG, RANKL, and sclerostin levels in the vegetarian and omnivorous children, significant associations were observed between sclerostin and CTX-I, as well as the P1NP/CTX-I ratio. This underscores the role of osteocyte-secreted sclerostin in regulating osteoblastogenesis through the inhibition of the Wnt/β-catenin signaling pathway [15].

There were no significant differences in the levels of regulatory markers, IGF-I and 25-hydroxyvitamin D, between the two groups. The similar concentrations of 25-hydroxyvitamin D can be explained by the fact that the majority (80%) of the participants (both vegetarians and omnivores) were taking vitamin D supplements (with an average dose of 600 ± 200 IU/day). In turn, IGF-I plays a role in bone physiology by promoting bone cell proliferation and differentiation, maintaining proper bone mass [53]. In our study, significant correlations of IGF-I with osteocalcin, CTX-I, and 25-hydroxyvitamin D suggest its involvement not only in bone formation but also in osteoclast activity.

Determining a panel of myokines, we observed comparable serum concentrations of myostatin, myonectin, and FGF-21, except for decorin, which showed significantly higher levels in the children following a vegetarian diet. The DCN/MSTN ratio was also higher in the vegetarians. The lack of a significant difference in the myostatin levels coincides with our previous study concerning myostatin and irisin, where the concentrations were similar in lacto-ovo-vegetarian and omnivorous children [54]. However, there are limited data concerning other myokines, such as decorin, myonectin, or FGF-21, in vegetarians.

Decorin, known for its effect on collagen fibrillogenesis, muscle tissue organization, and cellular signaling, acts as an antagonist of myostatin, potentially explaining the slightly lower myostatin levels observed in the vegetarians. Recent studies have suggested a positive role for decorin in bone metabolism, promoting osteoblast differentiation and inhibiting osteoclast activity [24,55].

Myonectin and FGF-21 are two myokines involved in glucose and lipid metabolism and are sensitive to nutrient changes. Their secretion is stimulated by feeding and inhibited by fasting [56]. Physical activity also increases the levels of these myokines. Myonectin acts via the Akt/mTOR system, a nutrient-responsive anabolic pathway, protecting against sarcopenia [57,58]. Contrary to expectations, we found no correlation between myonectin and the nutritional parameters.

The FGF-21 levels were inversely correlated with the dietary energy and protein intake, as well as with the anthropometric parameters. McCarty et al. [59] reported that a vegan diet, which is relatively low in protein and certain essential amino acids, may increase the hepatic activity of the kinase GCN2, consequently activating the liver’s production of FGF-21, a factor which favorably affects serum lipids. It has been proposed that plant proteins, owing to their amino acid composition, may promote FGF-21 activity. The proper quality of protein with sufficient essential amino acids, including branched-chain amino acids, is critical for muscle protein synthesis, which is also essential for maintaining adequate bone strength and density [60,61]. Our previous report showed that vegetarian children had a lower dietary intake and serum levels of valine, lysine, leucine, and isoleucine than meat-eaters [62]. These amino acids were more concentrated in animal-based protein compared to plant proteins, but it is difficult to explain individual dietary intake of nutrients because they act synergistically to maintain body homeostasis. In the present study, the vegetarian children had a lower dietary protein intake (but still within the reference range) than the omnivores, but there was no difference in the FGF-21 levels between the studied groups. We found relationships between this myokine and the dietary energy and protein intake, as well as anthropometric parameters. However, these associations may be related to children’s development, and after adjusting for age and gender, they are no longer statistically significant.

We observed that myokines significantly correlate with bone metabolism markers in children on a vegetarian diet. The strongest positive relations of serum myostatin with P1NP and sRANKL levels were observed in the simple correlation analysis and revealed in the multivariate regression model. This suggests a potential role for myostatin in bone metabolism regulation, particularly through the RANK/RANKL/OPG signaling pathway. Additionally, the positive association between myonectin and OPG observed in the simple correlation analysis indicates complex interactions between the skeletal and muscular systems, with implications for health and physiological functioning.

Interpreting myokine and bone metabolism marker levels in a clinical context is challenging due to limited research and reference values for healthy children and adolescents [63,64]. There are no the reference values for myokines and bone regulatory markers (RANKL, OPG, sclerostin) in the pediatric population. Moreover, myokines are produced not only in skeletal muscle but also in other tissues and organs, complicating their interpretation. Recently, Magaro et al. [65] discovered that sclerostin, primarily synthesized by osteocytes and known as an inhibitor of the Wnt-β catenin signaling pathway, affecting bone formation, is also produced by myoblasts, thereby functioning as a myokine. Their observation suggests that sclerostin released by skeletal muscle may interact synergistically with osseous sclerostin to regulate osteogenesis, possibly via a paracrine or local mechanism. In the present study, we found no differences in the sclerostin levels among the studied groups of children, nor any significant correlation with myokines. In the growing skeleton, the downregulation effect on osteogenesis by sclerostin from both osseous and muscular origins may be masked by the increase in body mass during growth. A deeper understanding of the molecular mechanisms responsible for the muscle–bone crosstalk is essential. Despite recent efforts, data on the molecular basis of this interaction remain limited.

Our study has several limitations. First, the cross-sectional design of this study restricts its ability to establish causality between dietary patterns, myokine levels, and bone metabolism markers. Longitudinal studies are needed to elucidate the long-term effects of vegetarian diets on muscle and bone health outcomes. Second, the relatively small number of participants limits the generalizability of our findings. Although we detected statistically significant associations, the sample size restricts our ability to explore a larger number of variables or their interactions in detail. However, we recruited the maximum number of prepubertal Polish healthy children who have been following a vegetarian diet since birth. Third, our results are based on single measurements of myokines and bone metabolism markers, which may not reflect the long-term exposure of these proteins. Nonetheless, we used reliable laboratory techniques to ensure the accuracy of our biochemical data and provided a unique panel of four myokines (myostatin, myonectin, decorin, and FGF-21) in prepubertal children on different diets for the first time. Fourth, while we did not perform an exact analysis of physical activity, both groups were comparable regarding PA intensity. Fifth, we assessed only the energy intake and the percentage of energy from protein, fat, and carbohydrates, but we plan to analyze dietary macro- and micronutrients (particularly amino acids) in relation to myokines in future studies. Including both vegetarian and omnivorous children captured the variability in dietary habits, enhancing the relevance and applicability of the findings. Finally, we did not perform densitometry scans but we studied healthy prepubertal children without a history of bone fractures. We plan to assess the bone mineral density as well as the body composition (fat mass, lean mass, bone mineral content) in adolescents on a vegetarian diet in the future.

Overall, our findings highlight the importance of considering dietary patterns and their potential impact on bone and muscle metabolism in children. Further research is warranted to elucidate the mechanisms driving the observed differences in bone markers and myokine levels between vegetarians and omnivores, as well as their implications for long-term musculoskeletal health in pediatric populations. Additionally, investigating dietary interventions or supplementation strategies to optimize bone health in children following vegetarian diets may have important clinical implications. We are convinced that this study provides valuable insights into the relationship between dietary patterns, myokines, and bone health in children, highlighting areas for further research and intervention development.

## 5. Conclusions

In conclusion, prepubertal vegetarian children exhibited impaired bone metabolism with reduced bone formation and increased bone resorption. Higher levels of decorin, a myokine involved in collagen fibrillogenesis and essential for tissue structure and function, may suggest a potential compensatory mechanism contributing to maintaining bone homeostasis in vegetarians. The observed positive correlations between myostatin and bone formation and resorption markers may suggest a complex interplay between muscle and bone metabolism, potentially mediated through the RANK/RANKL/OPG signaling pathway. Further investigations are important to fully understand the myokine-/bone-related mechanisms involved in muscle–bone crosstalk in relation to diet.

## Figures and Tables

**Figure 1 nutrients-16-02009-f001:**
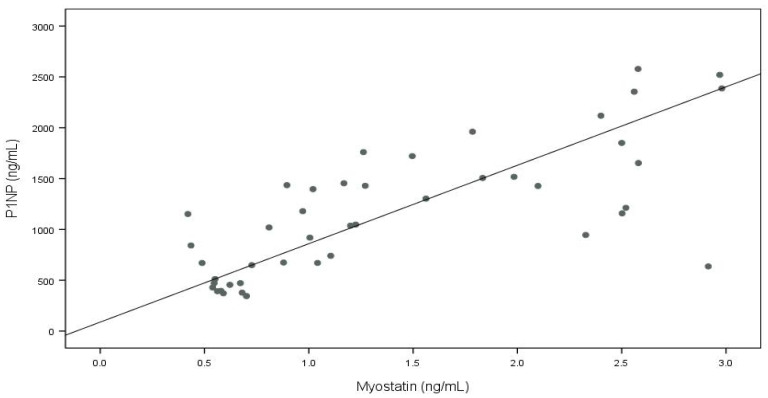
Serum myostatin and P1NP concentrations in vegetarian children (bivariate quantile regression).

**Figure 2 nutrients-16-02009-f002:**
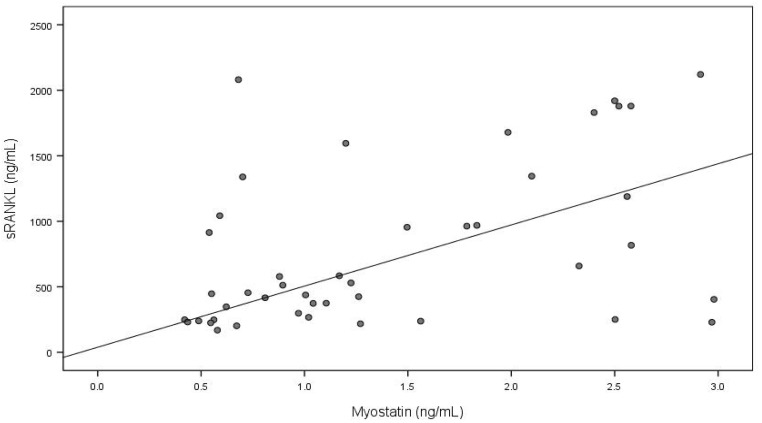
Serum myostatin and sRANKL concentrations in vegetarian children (bivariate quantile regression).

**Table 1 nutrients-16-02009-t001:** Anthropometric parameters and dietary intake of energy and percentage of energy from macronutrients in vegetarian and omnivorous children.

	Vegetarians (*n* = 44)	Omnivores (*n* = 24)	*p* Value
*n* (% boys)	22 (50)	13 (54)	0.743
Age (years)	6.3 (5.0–8.5)	6.0 (4.6–7.6)	0.406
Body weight (kg)	22.4 ± 6.5	20.6 ± 6.0	0.311
Body weight z-score	−0.50 ± 0.55	−0.54 ± 0.43	0.891
Height (cm)	120.5 ± 14.8	116.4 ± 13.7	0.134
Height z-score	−0.21 ± 0.96	−0.42 ± 0.74	0.443
BMI (kg/m^2^)	14.8 (14.2–15.9)	14.7 (14.3–15.6)	0.971
BMI z-score	−0.58 (−0.86–−0.19)	−0.58 (−0.76–−0.36)	0.602
Energy (kcal/d)	1392.0 ± 366.5	1534.6 ± 442.3	0.252
Protein, % of energy	12.1 ± 2.3	15.7 ± 2.7	<0.001
Protein (g/d)	41.2 ± 10.0	59.9 ± 19.8	<0.001
Fat, % of energy	30.2 ± 5.4	31.3 ± 4.3	0.778
Carbohydrates, % of energy	56.0 ± 5.3	51.8 ± 5.0	0.011
Dietary calcium (mg/d)	518.8 ± 174.7	617.3 ± 235.1	0.124
Dietary phosphorus (mg/d)	798.8 ± 219.4	895.5 ± 207.4	0.076
Dietary magnesium (mg/d)	224.8 ± 92.1	235.1 ± 84.2	0.541
Dietary vitamin D (µg/d)	1.27 (0.79–2.40)	1.99 (1.32–2.89)	0.079

Data are reported as percentage (%); mean ± standard deviation (SD) for normally distributed variables; median and interquartile ranges (IQRs) for skewed variables; BMI—body mass index.

**Table 2 nutrients-16-02009-t002:** Serum concentrations of bone metabolism markers and myokines in children on vegetarian and omnivorous diets.

	Vegetarians (*n* = 44)	Omnivores (*n* = 24)	*p* Value
Bone markers:			
P1NP (ng/mL)	1100 (639–1514)	1565 (1319–2485)	0.001
OC (ng/mL)	67.8 (56.0–93.6)	68.8 (56.7–101.5)	0.724
CTX-I (ng/mL)	1.911 ± 0.507	1.656 ± 0.345	0.018
P1NP/CTX-I ratio *	0.52 (0.33–0.95)	1.01 (0.81–1.29)	0.001
OC/CTX-I ratio	38.4 (31.2–54.6)	44.2 (34.7–69.6)	0.098
OPG (pmol/L)	4.63 (3.47–5.48)	4.51 (4.08–5.20)	0.590
sRANKL (ng/mL)	484 (255–1152)	490 (276–1156)	0.959
Sclerostin (ng/mL)	0.347 ± 0.095	0.354 ± 0.092	0.383
IGF-I (ng/mL)	151.5 (115.4–195.9)	154.0 (115.9–193.4)	0.928
25-OH D (ng/mL)	30.0 (27.0–33.7)	31.8 (28.3–37.0)	0.205
Myokines:			
MSTN (ng/mL)	1.14 (0.67–2.27)	1.48 (0.98–2.42)	0.267
Myonectin (ng/mL)	6.98 (4.79–8.65)	8.06 (6.36–8.90)	0.129
DCN (ng/mL)	84.9 ± 12.1	75.0 ± 16.9	0.020
FGF-21 (pg/mL)	133.0 (71.9–310.0)	138.8 (76.2–195.5)	0.833
DCN/MSTN ratio *	0.07 (0.04–0.13)	0.06 (0.03–0.08)	0.039

Data are reported as mean ± standard deviation (SD) for normally distributed variables and median and interquartile ranges (IQRs) for skewed variables; 25-OH D—25 hydroxyvitamin D; IGF-I—insulin-like growth factor-I; P1NP—*N*-terminal propeptide of type I procollagen; OC—osteocalcin; CTX-I—*C*-terminal telopeptide of collagen type I; OPG—osteoprotegerin; sRANKL—soluble receptor activator of nuclear factor kappa-B ligand; FGF-21—fibroblast growth factor 21; MSTN—myostatin; DCN—decorin; * Result was scaled by dividing by 1000.

**Table 3 nutrients-16-02009-t003:** Bivariate associations between myokine concentrations and age and anthropometric and nutritional parameters in vegetarian children.

	Myostatin		Myonectin		Decorin		FGF-21	
	rho *	*p*	rho *	*p*	rho *	*p*	rho *	*p*
Age	0.530	<0.001	−0.325	0.036	0.114	0.470	−0.362	0.016
Anthropometric parameters								
Weight	−0.473	0.001	−0.249	0.112	0.113	0.477	−0.381	0.011
Height	−0.491	0.001	−0.340	0.028	0.073	0.645	−0.317	0.036
BMI	0.023	0.883	0.042	0.794	0.141	0.373	−0.163	0.291
Nutritional parameters								
Energy (kcal)	−0.504	0.001	−0.178	0.284	0.073	0.664	−0.357	0.024
Protein%	−0.076	0.639	0.070	0.675	0.120	0.472	−0.241	0.133
Fat%	−0.242	0.132	−0.009	0.957	−0.022	0.898	0.155	0.329
Carbohydrate%	0.300	0.060	−0.073	0.663	0.211	0.204	−0.045	0.783
Protein (g/day)	−0.387	0.014	−0.117	0.486	−0.022	0.897	−0.393	0.012

* Spearman’s rho; FGF-21—fibroblast growth factor 21.

**Table 4 nutrients-16-02009-t004:** Bivariate associations between myokine concentrations and bone metabolism markers in vegetarian children.

	Myostatin		Myonectin		Decorin		FGF-21	
	rho *	*p*	rho *	*p*	rho *	*p*	rho *	*p*
P1NP	0.748	<0.001	0.277	0.075	−0.154	0.330	0.139	0.369
OC	−0.330	0.028	−0.017	0.916	0.147	0.353	−0.110	0.478
CTX-I	−0.046	0.765	0.061	0.701	0.223	0.156	−0.079	0.612
OPG	0.242	0.114	0.347	0.024	−0.026	0.871	−0.003	0.984
sRANKL	0.437	0.003	0.029	0.854	0.215	0.171	−0.143	0.355
Sclerostin	−0.286	0.059	0.162	0.305	−0.016	0.922	−0.026	0.867
IGF-I	−0.124	0.421	−0.200	0.204	0.081	0.609	−0.157	0.308
25-OH D	0.303	0.051	0.137	0.394	0.052	0.749	0.249	0.111
P1NP/CTX-I	0.636	<0.001	0.225	0.152	−0.191	0.226	0.090	0.561
OC/CTX-I	−0.334	0.027	−0.054	0.734	−0.022	0.891	−0.012	0.938

* Spearman’s rho; P1NP—*N*-terminal propeptide of type I procollagen; OC—osteocalcin; CTX-I—*C*-terminal telopeptide of collagen type I; OPG—osteoprotegerin; sRANKL—soluble receptor activator of nuclear factor kappa-B ligand; IGF-I—insulin-like growth factor-I; 25-OH D—25 hydroxyvitamin D; FGF-21—fibroblast growth factor 21.

**Table 5 nutrients-16-02009-t005:** Crude and age-adjusted associations between myokines and bone metabolism markers in vegetarian children (quantile regression).

	Crude				Age Adjusted			
	β	95% CI	*p*	Pseudo R^2^	β	95% CI	*p*	Pseudo R^2^
Myostatin								
P1NP	771	515; 1027	<0.001	0.390	810	462; 1159	<0.001	0.394
OC	−11.9	−26.2; 2.4	0.101	0.008	−0.4	−17.8; 17.1	0.967	0.031
sRANKL	467	214; 720	0.001	0.178	474	138; 810	0.007	0.185
Sclerostin	−0.011	−0.058; 0.035	0.629	0.018	−0.009	−0.065; 0.046	0.737	0.029
25-OH D	2.78	0.49; 5.06	0.018	0.076	2.85	−1.08; 6.79	0.151	0.097
P1NP/CTX-1	252.59	59.05; 446.13	0.012	0.165	251.15	−7.03; 509.33	0.056	0.165
OC/CTX-1	−4.9	−11.7; 1.8	0.148	0.057	−4.4	−13.6; 4.7	0.330	0.058
Myonectin								
P1NP	104	−28; 236	0.120	0.044	42	−97; 181	0.544	0.149
OPG	0.19	0.08; 0.46	0.159	0.084	0.19	−0.11; 0.48	0.209	0.085

P1NP—*N*-terminal propeptide of type I procollagen; OC—osteocalcin; CTX-I—*C*-terminal telopeptide of collagen type I; OPG—osteoprotegerin; sRANKL—soluble receptor activator of nuclear factor kappa-B ligand; IGF-I—insulin-like growth factor-I; 25-OH D—25 hydroxyvitamin D; FGF-21—fibroblast growth factor 21.

## Data Availability

The datasets generated for this study are available upon request to the corresponding author due to privacy.

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
