# Peer review of "Comparative Analysis of Myokines and Bone Metabolism Markers in Prepubertal Vegetarian and Omnivorous Children"

_nutrients, 2024, doi:10.3390/nu16132009_

Round 1
Reviewer 1 Report
Comments and Suggestions for Authors
This study provides valuable insights into the interplay between myokines, bone metabolism markers, and dietary patterns in prepubertal children. Its strengths include addressing a novel research question on the potential impact of vegetarian diets on the muscle-bone crosstalk, and the comprehensive analysis of multiple myokines and bone markers. However, the cross-sectional design, relatively small sample size, and lack of consideration for potential confounding factors limit the ability to establish causal relationships and the generalizability of the findings.
1. The study had a relatively small sample size, with only 44 vegetarians and 24 omnivorous children. This limited sample size may have reduced the statistical power to detect significant differences between the two groups, especially for some of the less pronounced effects.
2. The cross-sectional design of the study precludes the establishment of causal relationships between diet, myokine levels, and bone metabolism markers. While the results suggest associations, it is unclear whether the vegetarian diet directly influenced the observed differences in myokines and bone markers, or if other unmeasured factors may have played a role.
3. The study did not account for potential confounding factors that could influence muscle and bone health, such as physical activity levels, sun exposure (affecting vitamin D status), and pubertal stage.
4. While dietary intake data was collected, the study did not provide detailed information on the specific nutrient composition of the vegetarian and omnivorous diets, particularly regarding nutrients crucial for bone health, such as calcium, vitamin D, and protein.
5. While the results suggest vegetarian diets may be associated with altered bone metabolism, with reduced bone formation and increased resorption, the underlying mechanisms remain unclear.
6. Larger longitudinal or interventional studies with comprehensive assessments of nutrient intake, physical activity, clinical outcomes like bone mineral density, and other relevant factors are needed.
Author Response
Response to Reviewer 1:
Dear Reviewer,
Thank you for your thorough and insightful review of our manuscript. We appreciate your recognition of the novel research question and the comprehensive analysis we conducted. We acknowledge the limitations you pointed out and have addressed them in the revised manuscript. Here are our responses to your specific comments:
- Sample Size: We recognize that the relatively small sample size may have impacted our ability to detect significant differences between the groups and mentioned about this in the limitations. However, we recruited the maximum number of prepubertal Polish healthy children who have been following a vegetarian diet since birth. We plan to conduct further research on larger group of vegetarians, not only in the prepubertal age but also among adolescents. Nonetheless, we would like to point out that available studies on vegetarian children have similar sample sizes [Hovinen et al. EMBO Mol Med 2021, 13, e13492; Rowicka et al. Antioxidants 2023, 12, 682].

- Cross-Sectional Design: We agree that the cross-sectional study does not establish causal relationships between muscle and bone. We are aware that it is a complex mechanism dependent on many factors, including diet.

Proper quality of protein with sufficient essential amino acids, including branched-chain amino acids, is critical for muscle protein synthesis, which is also essential for maintaining adequate bone strength and density. Our previous report showed that vegetarian children had lower dietary intake and serum levels of valine, lysine, leucine, and isoleucine than meat-eaters [Ambroszkiewicz et al. Nutrients 2023, 15, 1376]. These amino acids are more concentrated in animal-based protein compared to plant proteins. Dietary protein intake, which is lower in vegetarians, along with a lower intake of essential amino acids, may affect muscle and bone metabolism. However, it is difficult to explain the role of individual nutrients because they act synergistically and multifunctionally.
- Potential Confounding Factors: Thank you for this comments. We acknowledge that factors such as physical activity levels, sun exposure, and pubertal stage could influence muscle and bone health.

All examined children were in prepubertal period. Physical activity levels in both groups of children were similar, as noted in the Material and Methods section. On average, participants engaged in about 60-90 minutes per day of moderate-to-vigorous physical activity (MVPA) and approximately 30 minutes per day of vigorous physical activity (VPA) daily, including regular activities after school.
A similar number of children in the vegetarian and omnivorous groups were studied during the autumn-winter period (October-April) and the spring-summer period (May-September) to avoid differences in vitamin D levels caused by varying sunlight exposure across different seasons. Specifically, 21 vegetarians (48%) were studied in the autumn-winter period and 23 vegetarians (52%) in the spring-summer period. For the omnivorous group, 12 children (50%) were studied in the autumn-winter period and 12 (50%) in the spring-summer period. Information regarding the seasonality of vitamin D has been added to the manuscript.
- Nutrient Composition: While we collected dietary intake data, we agree that providing detailed information on the specific nutrient composition, particularly for nutrients crucial for bone health, would enhance the study. So, we have now included more detailed data, such as dietary intake of calcium, phosphorus, magnesium, and vitamin D, in studied groups of children in Table 1. Thank for this valuable suggestion.

5. Underlying Mechanisms: We acknowledge that the mechanisms underlying the observed associations remain unclear. We have highlighted this in the discussion and emphasized the need for further research to elucidate these mechanisms. We suggested that although the mechanism is not fully understood, reduced bone formation may be related to collagen synthesis, which is evident in the decreased levels of P1NP, a marker of the early stages of bone formation. Lower protein intake together with its quality (reduced supply of branched-chain amino acids) may be associated with impaired bone and muscle metabolism.
- Future Studies: We fully agree with your suggestion that larger longitudinal or interventional studies with comprehensive assessments of nutrient intake, physical activity, clinical outcomes like bone mineral density, and other relevant factors are needed. Therefore, we recognize the necessity for further research involving larger groups of children and we plan assess bone mineral density, bone mineral content, and body composition (fat mass, lean mass, bone mineral content) in children and adolescents on a vegetarian diets in the future.

Thank you once again for your valuable feedback.
Best regards,
Jadwiga Ambroszkiewicz
Response to Reviewer 1:
Dear Reviewer,
Thank you for your thorough and insightful review of our manuscript. We appreciate your recognition of the novel research question and the comprehensive analysis we conducted. We acknowledge the limitations you pointed out and have addressed them in the revised manuscript. Here are our responses to your specific comments:
- Sample Size: We recognize that the relatively small sample size may have impacted our ability to detect significant differences between the groups and mentioned about this in the limitations. However, we recruited the maximum number of prepubertal Polish healthy children who have been following a vegetarian diet since birth. We plan to conduct further research on larger group of vegetarians, not only in the prepubertal age but also among adolescents. Nonetheless, we would like to point out that available studies on vegetarian children have similar sample sizes [Hovinen et al. EMBO Mol Med 2021, 13, e13492; Rowicka et al. Antioxidants 2023, 12, 682].

- Cross-Sectional Design: We agree that the cross-sectional study does not establish causal relationships between muscle and bone. We are aware that it is a complex mechanism dependent on many factors, including diet.

Proper quality of protein with sufficient essential amino acids, including branched-chain amino acids, is critical for muscle protein synthesis, which is also essential for maintaining adequate bone strength and density. Our previous report showed that vegetarian children had lower dietary intake and serum levels of valine, lysine, leucine, and isoleucine than meat-eaters [Ambroszkiewicz et al. Nutrients 2023, 15, 1376]. These amino acids are more concentrated in animal-based protein compared to plant proteins. Dietary protein intake, which is lower in vegetarians, along with a lower intake of essential amino acids, may affect muscle and bone metabolism. However, it is difficult to explain the role of individual nutrients because they act synergistically and multifunctionally.
- Potential Confounding Factors: Thank you for this comments. We acknowledge that factors such as physical activity levels, sun exposure, and pubertal stage could influence muscle and bone health.

All examined children were in prepubertal period. Physical activity levels in both groups of children were similar, as noted in the Material and Methods section. On average, participants engaged in about 60-90 minutes per day of moderate-to-vigorous physical activity (MVPA) and approximately 30 minutes per day of vigorous physical activity (VPA) daily, including regular activities after school.
A similar number of children in the vegetarian and omnivorous groups were studied during the autumn-winter period (October-April) and the spring-summer period (May-September) to avoid differences in vitamin D levels caused by varying sunlight exposure across different seasons. Specifically, 21 vegetarians (48%) were studied in the autumn-winter period and 23 vegetarians (52%) in the spring-summer period. For the omnivorous group, 12 children (50%) were studied in the autumn-winter period and 12 (50%) in the spring-summer period. Information regarding the seasonality of vitamin D has been added to the manuscript.
- Nutrient Composition: While we collected dietary intake data, we agree that providing detailed information on the specific nutrient composition, particularly for nutrients crucial for bone health, would enhance the study. So, we have now included more detailed data, such as dietary intake of calcium, phosphorus, magnesium, and vitamin D, in studied groups of children in Table 1. Thank for this valuable suggestion.

- Underlying Mechanisms: We acknowledge that the mechanisms underlying the observed associations remain unclear. We have highlighted this in the discussion and emphasized the need for further research to elucidate these mechanisms. We suggested that although the mechanism is not fully understood, reduced bone formation may be related to collagen synthesis, which is evident in the decreased levels of P1NP, a marker of the early stages of bone formation. Lower protein intake together with its quality (reduced supply of branched-chain amino acids) may be associated with impaired bone and muscle metabolism.

- Future Studies: We fully agree with your suggestion that larger longitudinal or interventional studies with comprehensive assessments of nutrient intake, physical activity, clinical outcomes like bone mineral density, and other relevant factors are needed. Therefore, we recognize the necessity for further research involving larger groups of children and we plan assess bone mineral density, bone mineral content, and body composition (fat mass, lean mass, bone mineral content) in children and adolescents on a vegetarian diets in the future.

Thank you once again for your valuable feedback.
Best regards,
Jadwiga Ambroszkiewicz

Reviewer 2 Report
Comments and Suggestions for Authors
Dear authors,
I read your manuscript with interest and I congratulate you for your work. A healthy diet is extremely important for everyone, but especially for children, and unfortunately few studies are conducted in this regard on vegetarian or vegan children. Since children under the age of 8-9 are a vulnerable population and the decision of a correct and healthy diet does not belong to them, the subject addressed in your study seems to me all the more important. 
You study demonstrated that children following vegetarian diet showed impaired bone metabolism with reduced bone formation and increased bone resorption. Also, a higher level of decorin, a myokine involved in collagen fibrillogenesis, potentially reflect a compensatory mechanism in response to a vegetarian diet in the study group. These may have implications in the child metabolism and will need further attention. 
Regarding the manuscript, it is well structured, the study group is sufficient, although it could be bigger. The results are well presented and supported by tables and graphs, and the discussions are comprehensive. I have nothing to suggest except the correction of some minor typographical errors.
Thank you.
Author Response
Response to Reviewer 2:
Dear Reviewer,
Thank you for your positive and encouraging feedback on our manuscript. We are glad to hear that you found our study well-structured and the topic important. We appreciate your comments on the significance of our findings and your recognition of the need for further research.
We acknowledge your point regarding the sample size and agree that a larger study group would strengthen the findings. We plan expanding the study group in our future research.
Additionally, we have revised the conclusions to be better supported by the results and corrected the minor typographical errors you pointed out.
Thank you for your kind words and constructive suggestions.
Best regards,
Jadwiga Ambroszkiewicz

Round 2
Reviewer 1 Report
Comments and Suggestions for Authors
The manuscript is suitable for submission to the Nutrients, as the revisions have adequately addressed the deficiencies.
Author Response
I am grateful for your insightful suggestion.
In answer to the question, the P1NP to CTX-I ratio was calculated for each participant, and the values of this ratio were computed for both study groups. Since the ratios were large numbers, the results were scaled by divided by 1000. This information has been added in the footnotes under Table 3 (changes in violet).
Thank you once again.
Best regards,
Jadwiga Ambroszkiewicz